# Mechanism of the Change in the Intestinal Microbiota of C-Strain *Spodoptera frugiperda* (Lepidoptera: Noctuidae) after an Interspecific Transference between Rice and Corn

**DOI:** 10.3390/microorganisms11102514

**Published:** 2023-10-08

**Authors:** Teng Di, Yongping Li, Guangzu Du, Yanyan He, Wenqian Wang, Yunfeng Shen, Jizhi Meng, Wenxiang Xiao, Guanli Xiao, Bin Chen

**Affiliations:** 1State Key Laboratory of Conservation and Utilization of Biological Resources of Yunnan, College of Plant Protection, Yunnan Agricultural University, Kunming 650201, China; dtn1020@163.com (T.D.); lyp9779@163.com (Y.L.); duguangzu1986@163.com (G.D.); heyanyan0505@outlook.com (Y.H.); wqwangts@163.com (W.W.); 2School of Agriculture, Yunnan University, Kunming 650500, China; 3Plant Protection and Quarantine Station of Baoshan City, Baoshan 678000, Chinabsxwx@live.cn (W.X.); 4College of Agronomy and Biotechnology, Yunnan Agricultural University, Kunming 650201, China

**Keywords:** *Spodoptera frugiperda*, host transfer, intestinal bacteria, physiological enzymes

## Abstract

*Spodoptera frugiperda* (J.E.Smith) (Lepidoptera: Noctuidae) was first found in 2019 in Yunnan, China, and it was characterized as a corn strain; it was also found on rice strains there, and it damages rice in China, but little is known about the effect of host plant transfer on the intestinal microbiota and the activities of detoxification enzymes in the C-strain (corn strain) *S. frugiperda*. The intestinal microbiota and the protective enzyme activity of *S. frugiperda* that were transferred from rice plants were assessed, and the fourth generation of insects transferred from corn were studied; the gene types of *S. frugiperda* that were transferred from rice plants were tested using mitochondrial *Tpi* gene sequences. The results showed that the intestinal microbiota in the C-strain *S. frugiperda* were changed after the host transference, and the diversity and richness of the intestinal bacterial communities of the *S. frugiperda* feeding on rice were significantly reduced after the transfer of the host from corn. The predominant species of intestinal bacteria of the *S. frugiperda* on rice transferred from corn were *Enterococcus* and *Enterobacter*, with relative abundances of 28.7% and 66.68%; the predominant species of intestinal bacteria of the *S. frugiperda* that were transferred from rice and feeding on corn were *Enterococcus* (22.35%) and *Erysipelatoclostridium* (73.92%); and the predominant species of intestinal bacteria of *S. frugiperda* feeding on corn was *Enterococcus*, with a relative abundance of 61.26%. The CAT (catalase) activity of the *S. frugiperda* transferred from corn onto rice from corn was reduced, the POD (peroxidase) activity was significantly increased after the transfer from corn, and no significant variations were found for the SOD (superoxide dismutase), CarE (carboxylesterase), and GST (glutathione S-transferase) activities of *S. frugiperda* after the host plant transfer. The results showed that after feeding on rice, the activities of CAT and POD in the in *S. frugiperda* body changed in order to resist plant secondary metabolites from corn or rice, but there was no significant change in the detoxification enzymes in the body. In summary, switching the host plant between corn and rice induced variations in the intestinal microbiota in C-strain *S. frugiperda* owing to the strain difference between the C-strain and the R-strain (rice strain), and this was consistent with the results of the activities of detoxification enzymes. The results indicat that changes in intestinal microbiota and physiological enzymes may be important reasons for the adaptive capacity of C-strain *S. frugiperda* to rice.

## 1. Introduction

*Spodoptera frugiperda* (J.E.Smith) (Lepidoptera: Noctuidae) is commonly known as the fall armyworm [1], and it is native to the tropical and subtropical regions of the Americas [2]. Because of its wide host range [3] and high reproductive and migratory abilities, it has posed a major threat to corn [4]. *S. frugiperda* has a wide range of hosts, with up to 353 species from 42 genera in 76 families, including corn and rice [3]. In the last century, it was reported that *S. frugiperda* had two biotypes due to differences in host preferences, namely the rice strain and the corn strain [5]. The corn strain prefers corn, sugarcane, and wheat, whereas the rice strain prefers rice and weeds [6]. The use of genotyping has confirmed that the *S. frugiperda* that invaded China was a corn strain [7]. In China, *S. frugiperda* mainly causes damage to corn. However, it completes its growth and reproduction by feeding on other plants, such as sugarcane, sorghum, rice, cabbage, certain weeds, and other plants, when there is a shortage of the main host, corn [8].

When insects feed on different host plants, their adaptability to different hosts differs in terms of their biological and behavioral indicators, such as the larval development period, survival rate, pupation rate, adult emergence rate, and adult egg production [9]. The contents of nutrients and secondary substances in the host plant affects the feeding preference and behavior of herbivorous insects which, in turn, affects their developmental process [10]. It was reported that *Ostrinia furnacalis* Guenée (Lepidoptera: Pyralidae) had a significantly longer developmental and average generation period after feeding on Jingke 968 corn filaments with higher contents of soluble sugar, soluble protein, polyphenol oxidase, and superoxide dismutase [11]. Studies have shown that plants use secondary metabolites to protect themselves [12,13]. When herbivorous insects feed on plants, the secondary metabolites stress the insects, affecting their growth and development [14].

At the same time, however, insects respond to plant stress by changing their own enzymatic activities to reduce the damage they suffer from secondary metabolites [15]. For example, *Oedaleus asiaticus* (Bey-Bienko) (Orthoptera: Acrididae) demonstrates lower detoxification enzyme activity after feeding on host plants with lower contents of secondary metabolites [16]. The insect intestine contains a large number of symbiotic bacteria which play an important role in an insect’s adaptation to its host and environment [17,18]. The composition of an insect’s intestinal microbes is mainly affected by the sex of the insect, food, pesticide resistance, temperature, CO_2_ concentration, light, and other aspects [19,20,21,22,23]. There are many species of intestinal microbes in *S. frugiperda* [24,25,26], Moreover, the bacterial composition and diversity of the intestinal microbes in *S. frugiperda* are different for different host plants [9]. Because *S. frugiperda* is a polyphagous pest, studying the community composition and diversity of its intestinal microbes after host conversion can provide a theoretical basis for understanding how polyphagous insects are able to feed on a variety of host plants.

In order to study the differences in feeding of *S. frugiperda* between its main host plant and other host plants and to explore the principles of these differences, population differences between groups of *S. frugiperda* feeding on corn and rice were analyzed by comparing the activities of the protective and detoxification enzymes of *S. frugiperda* after feeding on leaves of the two crops. The activities of the protective and detoxification enzymes of *S. frugiperda* after feeding on leaves of the two crops were compared, and the genotype differences between the two populations of *S. frugiperda* after continuous feeding for multiple generations were analyzed. The community composition and diversity of *S. frugiperda* intestinal microbes after host conversion were also studied. This study can provide a theoretical basis for the host adaptability of *S. frugiperda.*

## 2. Materials and Methods

### 2.1. Host Plants and Insects

Twenty-three Chujing rice (*Oryza sativa* L.) plants and eighteen Xuanhong corn (*Zea mays* L.) plants were selected as the host plants in this study. They were planted in greenhouses based on a random block design, with each plot measuring 3.0 m in length and 1.0 m in width and with three replications for rice and corn each.

Twelve adult *S. frugiperda* couples (mated 1 day after their emergence) were randomly selected from the laboratory corn-fed population, and they were then kept inn jointing-stage rice and trumpet-stage corn plots, respectively. Both populations of adult *S. frugiperda* were fed 10% honey water.

### 2.2. Strain Analysis for S. frugiperda Individuals Feeding on Rice and Corn

The insect genomic DNA (Deoxyribo Nucleic Acid) from the third instar larvae of the corn population and the fourth generation of the rice population was extracted using the CTAB (cetyltrimethyl ammonium nromide) method [27]. A single larva was placed in a 1.5 mL centrifuge tube and ground after it was precooled with liquid nitrogen. Then, 600 μL of CTAB lysate was added, mixed well, and bathed in water at a constant temperature of 65 °C for 30 min, with gentle mixing every 10 min. Next, 600 μL of phenol/chloroform/isoamylol (at a volume ratio of 25:24:1) was added, mixed well, and centrifuged at 12,000 r/min for 10 min. The supernatant was removed, and the addition, mixing, and centrifuging of 600 μL of phenol/chloroform/isoamylol was repeated one more time. A quantity of 600 μL of precooled isopropyl alcohol was then added to the supernatant, and this was mixed and centrifuged at 12,000 r/min for 10 min before the supernatant was removed, leaving the pellet. The pellet was washed by adding 70% precooled ethanol and centrifuged at 10,000 r/min for 5 min to discard the supernatant. The washing with ethanol was repeated twice. Finally, 50 μL of ddH_2_O was added to the pellet to dissolve the DNA.

The *Tpi* (Triosephosphate isomerase) gene of *S. frugiperda* was amplified via reference to the method described by Zhang [7] and Nagoshi [28]. The primers were synthesized by Shanghai Shenggong Biotechnology Co., Ltd. (Shanghai, China), and their sequences were as follows: *Tpi*-F: 5′-GGTGAAATCTCCCCTGCTATG-3′ and *Tpi*-R: 5′-AATTTTATTACCTGCTGTGG-3′. A polymerase chain reaction (PCR) amplification was performed in a volume of 50 µL with the following reaction components: 1.5 μL each of upstream and downstream primers, 5 μL of DNA template, 25 μL of PCR Mix, and 17 μL of ddH_2_O. The reaction conditions were as follows: pre-denaturation at 94 °C for 5 min, denaturation at 94 °C for 30 s, annealing at 48 °C for 30 s, extension at 72 °C for 1 min, 34 cycles, and extension at 72 °C for 10 min. A quantity of 2 μL of PCR product was taken for detection via electrophoresis using 1% agarose gel, and the PCR product and primers were sent to Sangon Biotech (Shanghai) Co., Ltd. (Shanghai, China) for sequencing.

The bidirectional sequencing results were stitched and proofread using DNAMAN (V9.0, San Ramon, CA, USA) software, the *Tpi* gene fragment sequences of *S. frugiperda* were obtained, and the representative sequences were selected by comparing the sequences of the repeated samples. Based on Nagoshi’s *Tpi* method [28], there were 10 sites of difference between the *Tpi* fragments from the rice and corn strains of *S. frugiperda*, and 7 or more of them were considered to be from the corn strain of *S. frugiperda.*

### 2.3. Intestinal Bacteria of S. frugiperda Fed on Rice and Corn

*Sample Processing*. The insects were switched from corn and fed on rice for the F1 (the first generation), F2 (the second generation), and F3 (the third generation) generations until the F4 (the fourth generation) generation. The fourth instar larvae of the F4 generation of *S. frugiperda* fed on rice were then switched to corn. Ten larvae were selected as one group for intestine dissections, and each treatment was repeated 3 times. After individual starvation in a sterile Petri dish for 24 h, the larval surface was sterilized twice using 75% alcohol for 1 min each time and rinsed in sterile water three times to remove the alcohol. Then, the intestines of *S. frugiperda* were dissected. The intestines were placed in 2 mL sterile centrifuge tubes, using labeled 1 tube per 10 larval intestines; then they were placed in liquid nitrogen for 10 min before they were transferred to a −80 °C freezer to be stored for DNA extraction for a high-throughput sequencing analysis.

For the 16S rRNA gene-based microbiome analyses, a PowerSoil^®^ DNA Isolation Kit (MO BIO Laboratories, Beijing, China) was used to extract intestinal DNA from *S. frugiperda*. The full length of the bacterial 16S rRNA was determined using a pair of bacterial 16S universal primers: 27F (5′-AGRGTTTGATYNTGGCTCAG-3′) and 1492R (5′-TASGGHTACCTTGTTASGACTT-3′). The of the polymerase chain reaction (PCR) was amplified using a total volume of 20 µL with the following reaction components: 5–50 ng/(×uL) of genome DNA, 1 μL each of 10 μM upstream and downstream primers, 0.4 μL of KOD FX Neo (TOYOBO), 10 μL of KOD FX Neo Buf (2×), 4 μL of 2 Mm dNTP, and ddH_2_O. The reaction conditions were as follows: pre-denaturation at 95 °C for 5 min, denaturation at 95 °C for 30 s, annealing at 50 °C for 30 s, extension 72 °C for 1 min/1 kb, 30 cycles, and finally, extension at 72 °C for 7 min. PCR products were detected using agarose gel electrophoresis, quantified using ImageJ (V8.0, National Institutes of Health, Bethesda, MD, USA) software according to the electrophoresis results, and mixed according to the amount of output data and the fragment size required for each sample. Recycling and purification were carried out using 0.8× magnetic beads. The PacBio platform was used for the high-throughput sequencing of the full length of the bacterial 16S rRNA gene (Biomarker Technologies, Beijing, China).

The original subreads were corrected to obtain the circular consensus sequencing (CCS) sequence (SMRT Link, version 8.0), and then LIMA (v1.7.0) software was used to identify the CCS sequences of different samples and remove chimeras (UCHIME, version 8.1) through the barcode sequence to obtain high-quality CCS sequences. Sequences at a level of 97% similarity were clustered (USEARCH, v10.0), and OTUs (Operational Taxonomic Units) were filtered, using 0.005% of the number of sequences sequenced as a threshold. Species were annotated using the Silva database and an RDP (Ribosomal Database Project) classifier confidence threshold of 0.8 (v2.2). PyNAST was used for multisequence comparisons, the phylogenetic tree was constructed via the neighbor-joining method using MEGAN5 software (V5, University of Tübingen, Tübingen, Germany), and a taxonomic dendrogram analysis was performed. In addition, alpha diversity analysis, beta diversity analysis, significant species difference analysis, etc., were performed using the BMKCloud Platform (Biomarker Technologies, Beijing, China).

### 2.4. Enzyme Activity

The fourth instar larvae of *S. frugiperda* feeding on corn and rice were used as a source for the test of enzyme activity. Five larvae were selected for one treatment, which was repeated three times. The enzyme activity was assayed using kits produced by the Nanjing Jiancheng Institute of Biological Engineering to establish the total protein content and the CAT (catalase), POD (peroxidase), SOD (superoxide dismutase), GST (glutathione S-transferase), and CarE (carboxylesterase) activities. Under ice-bath conditions, the samples were ground and mixed with 900 μL of normal saline and then centrifuged at 2500 r/min for 10 min. The supernatant was taken as the enzyme solution to be measured, and the enzyme activity in the *S. frugiperda* sample was assayed in reference to He [29].

### 2.5. Statistical Analysis

The data were analyzed using SPSS software, V.25.0 (SPSS Inc., Chicago, IL, USA). The data were analyzed for their homogeneity of variance and normality using the Leven and Shapiro–Wilk tests before statistical analyses. All data are presented as means ± standard errors. Statistical significance was set at *p* < 0.05. Figures were created using GraphPad prism version 7.0 (GraphPad Software, San Diego, CA, USA).

## 3. Results

### 3.1. Host Strain Analysis

After the sequencing results removed the low-quality sequences at both ends, a sequence comparison within the group found that the *Tpi* gene sequences of the same host were consistent, so only one was used for comparison. The lengths of the *Tpi* gene sequences of the *S. frugiperda* individuals that fed on corn and rice were 394 bp and 393 bp, respectively, which is consistent with the length expected length via electrophoresis detection (Figure 1A). A sequence alignment showed 99.50% similarity in the sequences of *Tpi* genes for the *S. frugiperda* fed using the two hosts. Comparing the *Tpi* gene sequences of the *S. frugiperda* fed on corn and rice with the *Tpi* gene sequences of the rice strain, it was found that there were only three site differences in the *Tpi* gene sequences of the *S. frugiperda* that fed on corn and rice (Figure 1B). Among the 10 haplotype sites used to distinguish between the corn strain and rice strain, the samples for the corn and rice strains were exactly the same as for the corn strain. This showed that the *S. frugiperda* fed on corn in the laboratory was a corn strain, and it remained a corn strain after four generations of feeding on rice.

### 3.2. Influence of Host-Plant-Switching on Intestinal Bacteria

Using the PacBio sequencing platform, the marker gene was sequenced using single-molecule real-time sequencing (SMRT Cell), and 188,795 CCS sequences were obtained after barcode recognition. An average of 12,586 CCS sequences were generated per sample, and the sequencing depth was higher than 0.99. The basic sequencing information is shown in Table 1. The readings were clustered at a similarity level of 97.0%, and a total of 9 phyla and 55 genera of intestinal bacteria were annotated.

The intestinal bacteria of *S. frugiperda* in five groups of samples were all annotated as Firmicutes, Proteobacteria, and Bacteroidetes, and the most abundant bacterial phyla were Firmicutes and Proteobacteria (Figure 2A,B). The relative abundance of the phylum Firmicutes ranked as Z (97.59%) > C (79.47%) > R1 (60.69%) > R3 (14.99%) > R2 (29.00%), and the relative abundance of the phylum Proteobacteria was R2 (84.96%) > R3 (70.97%) > R1 (39.28%) > C (3.38%) > Z (1.08%). Therefore, it was demonstrated that the corn-feeding population (C) and the population that was transferred to corn after four generations of rice feeding (Z) were dominated by the phylum Firmicutes. Although Firmicutes were the main microorganisms in the intestines of the larvae of the first generation of the population fed on rice (R1), the abundance of Proteobacteria was also higher. In addition, the dominant phylum in the larvae which were fed on rice for two and three generations (R2 and R3) was Proteobacteria.

At the species level, each treatment could cluster bacteria from *Enterococcus*, *Enterobacter*, *Muribaculaceae*, *Lactobacillus*, *Ligilactobacillus*, and *Paucibacter*. The diversity of treatment C at the genus level was higher than that of other treatments, and the four most abundant genera in C were *Enterococcus* (61.26%) > *Akkermansia* (8.43%) > *Muribaculaceae* (5.82%) > *Lactobacillus* (5.14%). In the R group (R1, R2, and R3), the two genera with the highest levels of abundance were *Enterococcus* (59.45%, 14.78%, and 28.97%) and *Enterobacter* (38.83%, 77.41%, and 66.68%), whereas in the Z group, the two genera with the highest levels of abundance were *Erysipelatoclostridium* (73.92%) and *Enterococcus* (22.35%). This shows that the dominant intestinal bacteria of the *S. frugiperda* larvae fed on corn were *Enterococcus*, whereas *Enterococcus* and *Enterobacter* were the main bacterial species in the larval intestines of the *S. frugiperda* fed on rice, and *Erysipelatoclostridium* was the main bacterial species in the corn-fed populations after they were switched from rice.

The rarefaction curve (Figure 2C) showed that the number of species extracted from C and Z was significantly higher than the number of species extracted from R (R1, R2 and R3), reflecting the greater richness of bacterial species in the intestines of the corn-fed *S. frugiperda* compared with that of the rice-fed population. The alpha diversity index (Figure 3) shows that the ACE index values of C (58.15) and Z (46.16) were significantly higher than those of R (18.94, 32.55, and 19.12) (*F* = 6.07, df = 4, 14, *p* < 0.05), and the Chao1 index showed that the bacterial communities of the *S. frugiperda* groups C (61.33) and Z (47.01) were significantly richer than that of group R (16.28, 25.00, 13.37) (*F* = 7.27, df = 4, 14, *p* < 0.05). It can be seen that the richness of the intestinal bacterial community of *S. frugiperda* fed on corn was significantly higher than that of the rice-fed larvae. There were no significant differences in the Simpson index and the Shannon index between the treatments (*F* = 0.25, df = 4, 14, *p* = 0.91; *F* = 1.19, df = 4, 14, *p* = 0.37), indicating that host conversion had no significant effect on the diversity and dominance concentration of the intestinal bacteria of *S. frugiperda*.

The results of the principal component analysis (PCA) showed (Figure 2D) that the contribution rates of the first principal component and the second principal component to the sample difference were 60.42% and 31.60%, respectively. The distance between the samples showed that the rice population and the corn population had a certain distance and that the similarity of the bacterial compositions of the three populations was high. Furthermore, it could be clearly distinguished from the corn population and the population that were transferred to corn after eating rice. A UPGMA (unweighted pair group method with arithmetic mean) hierarchical clustering analysis showed (Figure 2E) that the 15 samples of intestinal bacteria from *S. frugiperda* were clustered into two branches. The R1, R2, and R3 of the rice population were clustered into one, and there was a certain overlap between the samples, whereas the C and Z groups were clustered in the other branch, and there was no obvious overlap between the samples, indicating that the change in the host plant changed the composition of the intestinal bacterial community of *S. frugiperda* and that the intestinal bacterial community could remain relatively stable when the insects fed continuously on the same host.

### 3.3. Enzyme Activity of S. frugiperda Fed on Corn and Rice

The changes in the activities of the protective enzymes in the larvae of the *S. frugiperda* fed on rice and corn are shown in Figure 4, and there are significant differences. The catalase (CAT) enzyme activity of *S. frugiperda* larvae was higher in the corn population, the peroxidase (POD) enzyme activity was higher in the rice population, and the difference in the levels of superoxide dismutase (SOD) enzyme activity between the populations was not significant. The CAT enzyme activity in the larvae of the *S. frugiperda* that were fed on corn was 315.83 μ/gHb, which was significantly higher than the value of 197.00 μ/gHb (*t* = 3.361, df = 4, *p* = 0.028) for those fed on rice, and the CAT enzyme activity of the rice population was 1.60 times that of the corn population. The POD enzyme activity of the larvae of the corn-fed *S. frugiperda* was 42.04 μ/mprot, which was significantly lower than the value of 61.08 μ/mprot (*t* = −11.116, df = 4, *p* < 0.01) for the rice-fed population, and the POD enzyme activity of the corn-fed population was 1.45 times that of the rice-fed population. The SOD enzyme activity of the *S. frugiperda* larvae that fed on corn was 5.07 μ/mgprot compared with 6.70 μ/mgprot for those fed on rice, and the SOD enzyme activity of the *S. frugiperda* that fed on rice was higher than for those that fed on corn, but the difference was not significant (*t* = −1.849, df = 4, *p* = 0.138). In conclusion, compared with the *S. frugiperda* fed on corn, the rice-fed population demonstrated decreased CAT enzyme activity and increased POD enzyme activity.

The changes in the levels of activity of the detoxifying enzymes in the larvae of *S. frugiperda* that fed on corn and rice are shown in Figure 4. The levels of CarE activity of the *S. frugiperda* fed on corn and rice were 1.23 and 1.29 μ/mg prot, respectively, and there was no significant difference (*t* = −0.215, df = 4, *p* = 0.841). The GST activity of the *S. frugiperda* that ate corn was 55.80 μ/mprot, which was higher than the 44.58 μ/mprot of the rice-fed larvae, but there was no significant difference between the two hosts (*t* = 2.516, df = 4, *p* = 0.066). In conclusion, the host plants had no significant effect on the activity of detoxification enzymes in the larvae of *S. frugiperda.*

## 4. Discussion

The *S. frugiperda* that invaded China has a strong corn specialization and has caused serious harm to Chinese corn production [30,31], but it has also been reported to damage rice [32]. It was reported that *S. frugiperda* has two haplotypes: a corn strain and a rice strain [5]. Zhang et al. identified 82 samples invading Yunnan in China using *Tpi* haplotyping at the beginning of the invasion of *S. frugiperda* in China [7]. Nagoshi found that the *COI* gene sometimes deviated from the haplotype determination of *S. frugiperda* [28], so the *Tpi* gene fragments located in the nuclear genome are mainly used for identification. Based on an analysis of mitochondrial *Tpi* gene sequences, our experiment found that the laboratory corn population was a corn strain, and the same gene type was also found after four generations of feeding on rice.

By studying the diversity and composition of the intestinal bacterial community of *S. frugiperda* when the insects were switched between rice and corn, it was found that in the process of switching from corn to rice and then to corn, the number of OTUs in the intestines of *S. frugiperda* showed a trend of decreasing and then increasing. The dominant phylum of the corn population is Firmicutes, while the dominant phyla of the rice population include Firmicutes and Proteobacteria. The dominant genus of bacteria in the corn population is *Enterococcus*, while the dominant phyla of bacteria in the rice population include *Enterococcus* and *Enterobacter*. The dominant genus of bacteria in the conversion of the rice population into a corn population is *Erysipelatoclostridium*. We found that the dominant bacteria in the intestinal bacterial community of *S. frugiperda* were Firmicutes and Proteobacteria, which is consistent with other *S. frugiperda* populations [33,34].

In order to resist the harm of herbivorous insects, there are a large number of secondary metabolites in plants which mainly include phenylpropanoids, quinones, flavonoids, tannins, terpenoids, steroids and their glycosides, and alkaloids. There is also a large number of secondary metabolites in maize and rice, with significant differences in their contents. Phytophagous insects reduce the stress effect of plant secondary metabolites through a series of defensive behaviors, such as altering the levels of activity of physiological enzymes in the body [35]. The superoxide dismutase in insects can catalyze the dismutation of superoxide ion free radicals, peroxidase can eliminate hydrogen peroxide and toxic substances in animals and plants and participate in the metabolism of reactive oxygen species, and catalase can catalyze excess H_2_O_2_ into water and oxygen, providing protection for insects [36]. The level of CAT enzyme activity of the *S. frugiperda* which were fed on rice after being transferred from corn was significantly lower than the CAT enzyme activity of those fed on corn, and the POD enzyme activity was significantly higher than that of the corn population; this shows that feeding *S. frugiperda* with different host plants changed the activities of their enzymes, which may have been due to the different concentrations of secondary metabolites in the host plants. These are the same as the results achieved by Lu [37]. The diversity and richness of the bacterial communities in the intestines of the corn-fed population were significantly higher than those of the rice-fed population. The dominant genera of intestinal bacteria in the *S. frugiperda* fed on corn and rice were *Enterococcus* and *Enterobacter*, respectively. These results show that the corn strain of *S. frugiperda* is less adaptable to rice than to corn.

## 5. Conclusions

The transfer between host plants changed the composition of the intestinal bacteria and the activities of enzymes in *S. frugiperda* due to the different secondary metabolites in the plants. These establish a theoretical basis for designing different strategies for controlling strategies of *S. frugiperda* for different host crops. Also, this study may provide a theoretical basis for controlling *S. frugiperda* via maize–corn intercropping.

## Figures and Tables

**Figure 1 microorganisms-11-02514-f001:**
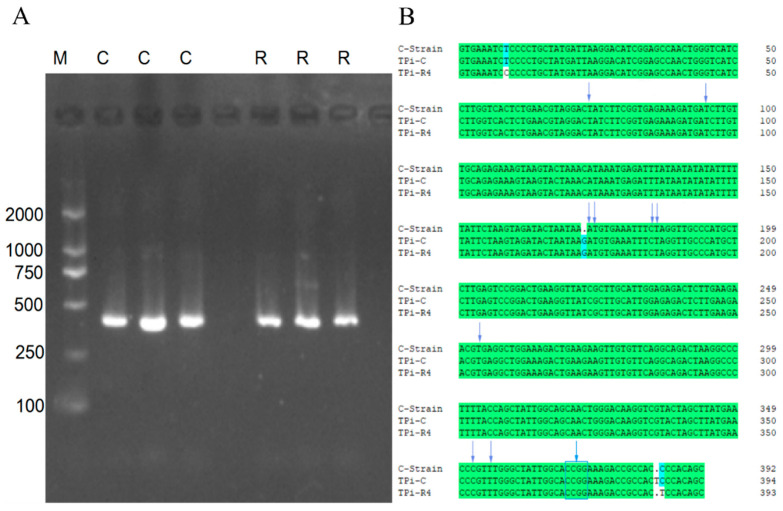
DNA extraction and genotype identification of *S. frugiperda*. (**A**) PCR amplification product agarose gel electrophoresis detection diagram; M represents DL1000 DNA marker; C represents corn population; R represents rice population. (**B**) Comparison of haplotype sites of *S. frugiperda* based on *Tpi* gene fragments; arrows indicate 10 polymorphic haploid loci for distinguishing corn subtypes from rice subtypes; the box represents the four gene loci of the 10th locus.

**Figure 2 microorganisms-11-02514-f002:**
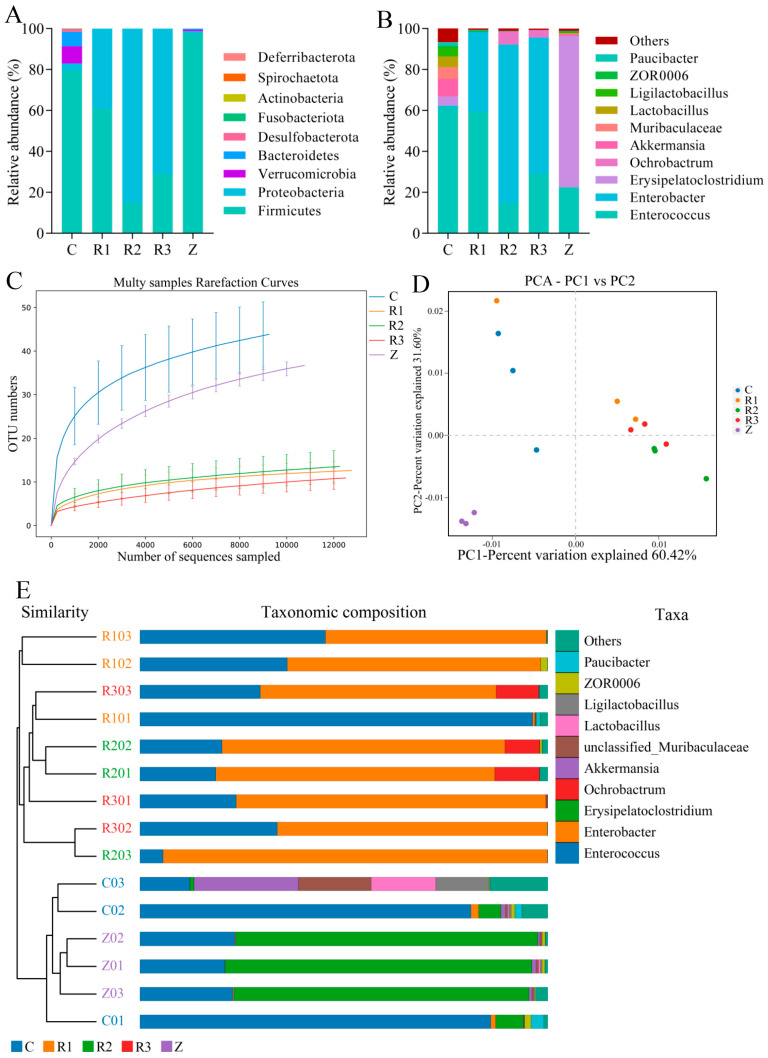
Parameters of the intestinal bacteria of *S. frugiperda*. (**A**) Phylum−based proportional composition of the intestinal bacteria of larvae. (**B**) Genus−based proportional composition of the intestinal bacteria of larvae. (**C**) Rarefaction curve for each sample. (**D**) Principal component analysis of the intestinal bacteria of *S. frugiperda*. (**E**) UPGMA analysis graph of the intestinal bacteria of *S. frugiperda*. C represents *S. frugiperda* fed on corn. R1, R2, and R3 represent the first, second, and third generations of *S. frugiperda* fed on rice, respectively. Z represents the F4 generation of the *S. frugiperda* fed on rice and then transferred to corn.

**Figure 3 microorganisms-11-02514-f003:**
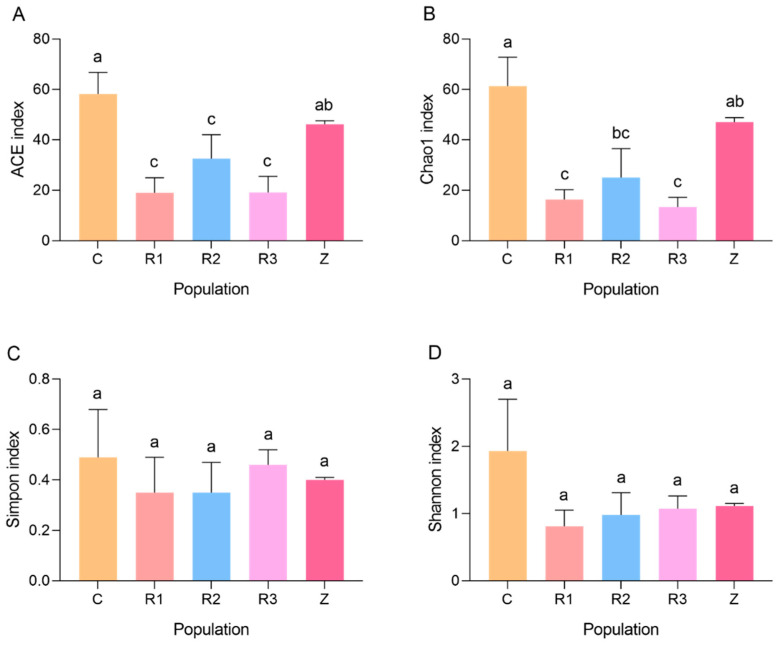
Alpha diversity index of the intestinal bacterial community of *S. frugiperda.* (**A**) ACE index of the intestinal bacterial community of *S. frugiperda.* (**B**) Chao1 index of the intestinal bacterial community of *S. frugiperda.* (**C**) Simpon index of the intestinal bacterial community of *S. frugiperda.* (**D**) Shannon index of the intestinal bacterial community of *S. frugiperda.* C represent *S. frugiperda* fed on corn. R1, R2, and R3 represent first, second, and third generations of *S. frugiperda* fed on rice, respectively. Z represents the fourth generation of *S. frugiperda* whose host was transferred to maize. Different lowercase letters indicate significant differences among the treatments at *p* < 0.05 (Tukey’s HSD test).

**Figure 4 microorganisms-11-02514-f004:**
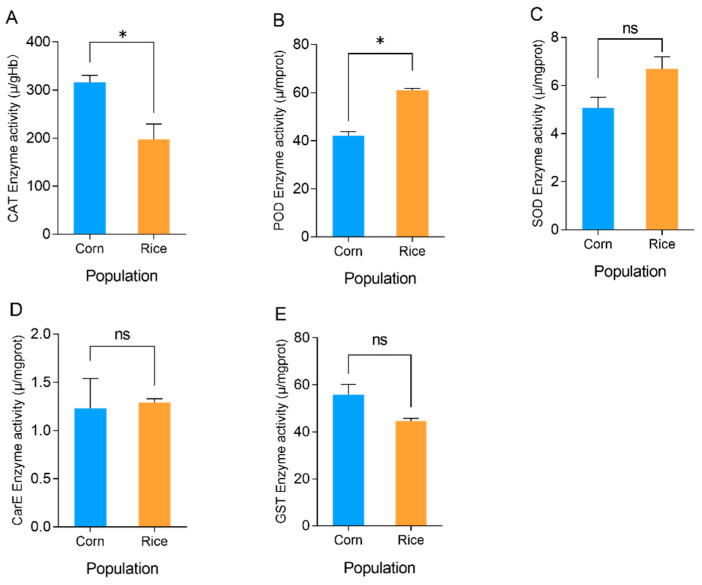
Enzyme activities of *S. frugiperda* fed on rice and corn (* *p* < 0.05; *t* test). (**A**) CAT (catalase) enzyme activity of *S. frugiperda*. (**B**) POD (peroxidase) enzyme activity of *S. frugiperda*. (**C**) SOD (superoxide dismutase) enzyme activity of *S. frugiperda*. (**D**) CarE (carboxylesterase) enzyme activity of *S. frugiperda*. (**E**) GST (glutathione S-transferase) enzyme activity of *S. frugiperda*.

**Table 1 microorganisms-11-02514-t001:** Basic information on the intestinal bacterial sequencing of *S. frugiperda* (Note: C represents the corn population; R1 represents the rice population of the first generation; R2 represents the rice population of the second generation; R3 represents the rice population of the third generation; Z represents the rice population of the fourth generation transferred to feeding on corn).

Group Information	Sample ID	Effective CCS	AvgLen (bp)	Tags Num	Goods Coverage
C	C01	12,913	1484	12,339	0.9993
	C02	10,909	1483	9274	0.9987
	C03	10,626	1474	10,000	0.9992
average		11,483	1480	10,538	0.9991
R1	R101	12,991	1483	12,930	0.9998
	R102	12,844	1471	12,784	0.9995
	R103	12,880	1473	12,821	1.0000
average		12,905	1476	12,845	0.9998
R2	R201	12,833	1461	12,641	0.9999
	R202	12,755	1464	12,466	0.9994
	R203	12,768	1467	12,689	0.9997
average		12,785	1464	12,599	0.9997
R3	R301	12,879	1470	12,779	0.9998
	R302	13,011	1472	12,879	0.9998
	R303	12,817	1463	12,605	0.9995
average		12,902	1468	12,754	0.9997
Z	Z01	12,311	1463	11,894	0.9991
	Z02	12,893	1464	12,525	0.9992
	Z03	11,383	1464	10,906	0.9990
average		12,196	1464	11,775	0.9991

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
