# Peer review of "Mechanism of the Change in the Intestinal Microbiota of C-Strain Spodoptera frugiperda (Lepidoptera: Noctuidae) after an Interspecific Transference between Rice and Corn"

_microorganisms, 2023, doi:10.3390/microorganisms11102514_

Round 1
Reviewer 1 Report
The manuscript "Reconstruction and mechanism of the intestinal microbiota in C-strain Spodoptera frugiperda (Lepidoptera: Noctuidae) after the interspecific transference between rice and corn" is good work but it needs some improvements in the English language and discussion section. The discussion section is too short and not according to results. My other comments and suggestions are given on manuscript attached.

English editing of manuscript is highly required.
Author Response
Responses to the referees’comments
Dear editors, reviewers
Thank you very much for your professional review and critical comments, which have further improved the quality of our manuscript. We have revised the manuscript according to your comments, and answered and marked them in the Word file for your review.
- Which year?(line 19)
Response: we have revised it.
- Rephrase sentence, poor English(line22-25)
Response: we have revised the sentence.
- It is very difficult to understand, Rephrase results and write clearly.(line27-32)
Response: we have revised the sentence.
- Explain CAT, POD, SOD, RDP, GST, CarE, etc. (lines 32-36, 167, 314, 316, etc.)
Response: we have provided a section with an abbreviation list at the end of the text.
- Provide authority name(line73)
Response: we have Provided authority name.
- Provide complete rearing protocol.(line100-102)
Response: We have revised the experimental workflow in the paper
- Provide complete taxonomic information(line239-245)
Response: we have revised the sentence.
- Why figure 1 is given at last of results??(line322-323)
Response: we have revised it.
In addition, we add a fifth part - Conclusion, to themanuscript, according to the editor’s recommendence.

Reviewer 2 Report
The submitted article aims to characterize changes in the microbiome and activity of antioxidant systems in the corn strain of Spodoptera frugiperda when transferring to rice plants and returning insects back to the initial host after four generations fed on rice. The authors demonstrated a significant decrease in microbiota richness and content as well as alterations in dominant species accordingly. The obtained results partially explained the adaptability of the studied strain on the microbiome level, which might provide insights into developing new strategies for pest biocontrol. Still, despite statistical rigor and valid experiments, the text cannot be recommended for publication, as in its current form it lacks essential details. Below please find my comments and suggestions.
1) It seems that the title misses the word change or alteration. The microbiota itself is not a process or mechanism.
2) The same goes for reconstruction. It is more suitable to use more straightforward synonyms. The word reconstruction is also applied to pangenomes, phylogeny, microbiome, etc. I suggest rephrasing it in the title and abstract (lines 25-26, 39). Moreover, stating that the results showed that microbiota was reconstructed (lines 25-26) does not imply the result but the process of reconstructing the microbial taxa abundance. Therefore, the results show changes, but not reconstruction.
3) The experimental workflow is not clear in the abstract. Please state that transfer from rice plants was performed for the 4th generation of insects transferred from corn. For now, it is unclear from the text.
4) In the Results section, the most abundant genus after the backward transfer was Erysipeloxlostridium but not Erysiplelothrix as was mentioned in the abstract (line 31).
5) The abstract should include the interpretation of changes in the activities of detoxification systems. Furthermore, the authors should explain why it is considered consistent with microbiota content. Please, provide a more detailed explanation (line 37).
6) Please, explicitly state in the abstract that C-strain is corn strain.
7) Please, refrain from using phrases such as American scientists and consider rephrasing the sentence (line 49).
8) The introduction section contains examples of other insect species. Please, add studies dedicated to Spodoptera frugiperda.
9) All the references in the text are badly formatted and do not meet the journal requirements (they should be numbered).
10) The word content or concentration seems to be missed (line 74).
11) What was the age of the plants used in the study? Please, describe.
12) The authors frequently do not spell out abbreviations and names, e.g., tpi, OTU, RDP, GST, CarE, etc. (lines 128, 167, 314, 316, etc.) Please, decipher all abbreviations when first mentioned. Moreover, I suggest providing a section with an abbreviation list at the end of the text.
13) For some programs, the exact versions are presented, while for others they are absent, please check and add the information.
14) The used software is not cited, and the respective section in the methods does not contain citations.
15) Please, state which utilities/packages/libraries were used for calculateing alpha diversity and other measurements.
16) Please, specify the version of R language but not only Rstudio.
17) Something is wrong with figure captions. Figure 4 is marked as Figure 1. Figure 6 mentioned in the text (line 230) does not exist.
18) Species names are not italicized in some places (lines 239-241), please carefully check the manuscript for correct italics.
19) Typo error (fedd, line 295).
20) The discussion is too scarce if exists. The authors simply describe the obtained results but do not discuss them. More attention should be paid to the comparative analysis. e.g., with other insect species or Spodoptera-related articles in terms of the composition of the microbiota, changes in detoxification enzyme activities, relationships between these observations, etc. The discussion, thus, requires substantial revision and should be more comprehensive. The authors should propose or describe mechanisms, perform comparative analysis, and make conceptual statements instead of descriptions only.
21) The concluding passage with perspectives and future directions of studies should be added.
Author Response
Responses to the referees’comments
Dear editors, reviewers
Thank you very much for your professional review and critical comments, which have further improved the quality of our manuscript. We have revised the manuscript according to your comments, and answered and marked them in the Word file for your review.
- It seems that the title misses the word change or alteration. The microbiota itself is not a process or mechanism.
Response: We agree with the suggestion, and we have added the word “change”.
- The same goes for reconstruction. It is more suitable to use more straightforward synonyms. The word reconstruction is also applied to pangenomes, phylogeny, microbiome, etc. I suggest rephrasing it in the title and abstract (lines 25-26, 39). Moreover, stating that the results showed that microbiota was reconstructed (lines 25-26) does not imply the result but the process of reconstructing the microbial taxa abundance. Therefore, the results show changes, but not reconstruction.
Response: Thanks for your good suggestion, and we replaced the word “reconstruction” with“change”. Moreover, we have rephrased it in the title and abstract.
- The experimental workflow is not clear in the abstract. Please state that transfer from rice plants was performed for the 4th generation of insects transferred from corn. For now, it is unclear from the text.
Response: We have revised the experimental workflow in the abstract.
- In the Results section, the most abundant genus after the backward transfer was Erysipeloxlostridiumbut not Erysiplelothrix as was mentioned in the abstract (line 31).
Response: Thanks for your careful review, we have revised it.
- The abstract should include the interpretation of changes in the activities of detoxification systems. Furthermore, the authors should explain why it is considered consistent with microbiota content. Please, provide a more detailed explanation (line 37).
Response: we have made a supplement on the interpretation of changes in the activities of detoxification systems in the abstract. Furthermore, we have provide a more detailed explanation for the consistent of enzyme activity with microbiota content.
- Please, explicitly state in the abstract that C-strain is corn strain.
Response: We have explicitly stated in the abstract that C-strain is corn strain.
- Please, refrain from using phrases such as American scientists and consider rephrasing the sentence (line 49).
Response: we have revised the sentence.
- The introduction section contains examples of other insect species. Please, add studies dedicated to Spodoptera frugiperda.
Response: we have revised it according to the reviewer’s suggestions.
- All the references in the text are badly formatted and do not meet the journal requirements (they should be numbered).
Response: we have revised according to the journal requirements.
- The word content or concentration seems to be missed (line 74).
Response: we have added the word “content”.
- What was the age of the plants used in the study? Please, describe.
Response: we have described the age of the plants used in the study.
- The authors frequently do not spell out abbreviations and names, e.g., tpi, OTU, RDP, GST, CarE, etc. (lines 128, 167, 314, 316, etc.) Please, decipher all abbreviations when first mentioned. Moreover, I suggest providing a section with an abbreviation list at the end of the text.
Response: we have provided a section with an abbreviation list at the end of the text.
- For some programs, the exact versions are presented, while for others they are absent, please check and add the information.
Response: we have checked and revised.
- The used software is not cited, and the respective section in the methods does not contain citations.
Response: we have described the software in the MM section.
- Please, state which utilities/packages/libraries were used for calculateing alpha diversity and other measurements.
Response: we have stated the packages were used for calculateing alpha diversity.
- Please, specify the version of R language but not only Rstudio.
Response: we have revised it.
- Something is wrong with figure captions. Figure 4 is marked as Figure 1. Figure 6 mentioned in the text (line 230) does not exist.
Response: we have checked and revised the Figure 4, deleted the word “Figure 6AB” in the text.
- Species names are not italicized in some places (lines 239-241), please carefully check the manuscript for correct italics.
Response: we have checked and revised the Species names in italicized.
- Typo error (fedd, line 295).
Response: we have revised it.
- The discussion is too scarce if exists. The authors simply describe the obtained results but do not discuss them. More attention should be paid to the comparative analysis. e.g., with other insect species or Spodoptera-related articles in terms of the composition of the microbiota, changes in detoxification enzyme activities, relationships between these observations, etc. The discussion, thus, requires substantial revision and should be more comprehensive. The authors should propose or describe mechanisms, perform comparative analysis, and make conceptual statements instead of descriptions only.
Response:We have revised the discussion according to the reviewers’ suggestions.
- The concluding passage with perspectives and future directions of studies should be added.
Response: we have made a supplement for the perspectives and future directions of studies.
In addition, we add a fifth part - Conclusion, to themanuscript, according to the editor’s recommendence.

Round 2
Reviewer 1 Report
The manuscript is much improved but still has some minor corrections. Comments are given on the pdf attached.

Minor language editing required
Reviewer 2 Report
The authors have addressed most of my comments; however, there are still some minor issues that should be considered before recommending the article for publication.
1. It is redundant to state “The abbreviation for” when spelling out the abbreviations. It, thus, should be omitted;
2. I suggest adding a possible explanation of the changes in detoxification enzyme activity (lines 42-43). This may reflect adaptation to different content of secondary metabolites produced by host plants as the authors mention in the Conclusion section;
3. Please, check the text for missed spaces, especially before citations;
4. Please, increase the font in figures 2C-D. The text in the legends is not visible;
5. The discussion section requires more details. It would be more informative for the reader to see the differences and similarities between the obtained observations and existing studies if specific details are provided. I suggest mentioning dominant genera in the microbiota as well as prevalent secondary metabolites in plants that can exert an effect on insects that are mentioned in the cited studies.
